# Characteristics and mechanisms of near-surface atmospheric electric field negative anomalies preceding the 5 September, 2022, Ms6.8 Luding earthquake, China

Lixin Wu[1, 2], Xiao Wang[1, 2], Yuan Qi[1, 2] *, Jingchen Lu[1, 2], Wenfei Mao[1, 2]

[1]School of Geosciences and Info-Physics, Central South University, Changsha, 410083, China

[2]Laboratory of Geo-Hazards Perception, Cognition and Predication, Central South University, Changsha, 410083, China

Corresponding author: Yuan Qi (weloveqy@163.com)

**Abstract.** A magnitude 6.8 strike-slip earthquake (EQ) struck Luding, Sichuan province, China, on September 5, 2022, resulting in significant damages to the nearby Ganzi Prefecture and Ya'an City. In this research, near-surface atmospheric electric field (AEF) recorded at four sites 15d before the Luding EQ were analyzed and discriminated, and multi-source auxiliary data including precipitation, cloud base height and low cloud cover were used at the same time. Nine possible seismic AEF anomalies at four sites were obtained preliminarily. Accordingly, microwave brightness temperature (MBT) data, which is very sensitive to the surface dielectrics and closely related to the air ionization, together with surface soil moisture, lithology, and 3D-simulated crustal stress field, was jointly analyzed to confirm the seismic relations of the obtained negative AEF anomalies. The geophysical environment for crustal high-stress concentration, positive charge carriers transfer and surface accumulation was demonstrated to exist and meet the conditions necessary for generating local negative AEF anomalies. Furthermore, to deal with the spatial disparities in sites and regions with potential atmospheric ionization, near-surface wind field data was employed to scrutinize the reliability of the AEF anomalies by comprehensively analyzing the spatial relationships among surface charges accumulation areas, wind direction and speed, as well as the AEF sites. Finally, four negative AEF anomalies were deemed to be closely related to the Luding EQ, and the remaining five possible anomalies were ruled out. A possible mechanism of negative AEF anomalies before Luding EQ is supposed to be that positive charge carriers were generated from the underground high stress concentration areas, and then transferred to and accumulated on the ground surface and to ionize the surface air, thus disturbing the AEF above the ground. This study presents a method for identifying and analyzing seismic AEF anomalies and is also beneficial for examining the pre-earthquake coupling process between the coversphere and the atmosphere.

**Keywords:** atmospheric electric field, seismic anomaly, P-holes, microwave brightness temperature, crustal stress field alteration

## 1 Introduction

In nature, the Global Electric Circuit (GEC) is driven by global thunderstorm activity and large-scale ion separation in charged cloud (Rycroft et al., 2000). In the background of GEC, a direct current (DC) atmospheric electric field with an amplitude of around 130 V/m is always present in undisturbed fair areas (Sun,1987). This electric field, also known as the fair-weather atmospheric electric field (FW-AEF), is oriented vertically downwards, which means that the atmosphere is positively charged relative to the Earth, while the Earth carries a negative charge (Li et al., 2022). In recent decades, some scientists have discovered that seismic activity can cause AEF anomalies with its direction opposite to FW-AEF in the seismogenic region. In 1966, Kondon (1966) claimed pre-earthquake (EQ) abnormal electric field signals by using the field mill instrument for the first time at the Matsushiro Observatory in Japan. Based on the electric field data recorded by Pixian site and Wenjiang site (in Chengdu, China), significant abnormal phenomena of AEF before the 2008 Ms8.0 Wenchuan EQ were found when the interference of lightningactivities were excluded (Li et al., 2017). Chen et al. (2022) also observed the AEF anomalies before the 2021 Ms4.3 Luanzhou EQ at two sites in Baodi and Yongqing, China. By analyzing the meteorological data, the anomalous signal monitored

at Baodi station was found to be influenced by a combination of transit clouds and geological activity, while the bay-type persistent electric field anomaly monitored at Yongqing station was considered as a possible AEF precursor of the EQ.

At present, there are three acceptable mechanisms for AEF anomalies before EQs. Firstly, it is considered that seismic-related anomaly in radon emanation can be linked to preseismic electromagnetic phenomena such as the great changes of small ion concentration and AEF (Omori et al., 2007). In recent study (Jin et al., 2020), the AEF reduction before the Wenchuan EQ was interpreted from the perspective of the rapid changing of radon concentration as the mainshock approaching. Besides, by combining the time series and dynamic periodogram of AEF anomalies from 6 hours before to 6 hours after the EQ, Hobara et al. (2022) attributed the phenomenon to the internal gravity waves generated near the epicenter passing through the AEF site, which changed the space charge density in the surface layer of the atmosphere. In addition, during the experimental study, Freund (2000, 2007, 2010) found that stress-activated carriers, named as P-holes, activated in the igneous and metamorphic rocks, are able to transfer along stress gradient and accumulate on the rock surface in unstressed areas or even on the ground surface covered by sands. When the P-holes arrive at the air-ground interface, a positive potential could be produced and air particles here are able to be ionized so as to change the near-surface AEF when it reached a high level (Freund, 2013). Meanwhile, the accumulation of P- holes on ground surface was also believed to reduce the surface microwave dielectric constant and enhance the regional microwave radiation (Mao et al., 2020; Qi et al., 2021a, 2021b).

Some other researchers have also proposed different opinions on the pre-EQ AEF anomalies observed at ground sites. Based on the statistical analysis of 103 pre-EQ bay-type AEF anomalies in the Kamchatka region, Smirnov et al. (2019) found that the duration and magnitude of AEF anomalies in hour-scale did not depend on either the magnitude of the EQ or the distance to the epicenter, while that in day-scale were related to the magnitude of the EQ. Hao et al., (1988) analyzed the AEF at the three stations in Baijiatong, Baodi and Beidaihe for several seismic events happened in and around Beijing from 1977 to 1986, and found that there were evident negative anomalies of AEF variation before the EQs, decaying significantly with the distance to the epicenter and being only associated with nearby EQs but not far strong EQs. However, most of the researches were based on statistical judgements and have not yet integrated with the regional geological conditions of seismogenious zone as well as the local crustal stress field alteration (CSFA), which is crucial to whether charges from the stressed rock mass of Earth's crust can ionize the near-surface atmosphere.

It's well known that atmospheric vertical electric field acts as a bridge connecting the surface charges and atmospheric particle concentration. The current consensus is that the increased concentration of atmospheric ions at the ground- air interface leads to the formation of additional vertical electric fields, which further transport ions from the lower atmosphere to the upper atmosphere, ultimately causing atmospheric anomalies. In our recent research, the multi-parameter seismic anomalies before the 2015 Nepal EQ sequence were analyzed systematically (Wu et al., 2023) by referring to the lithosphere–coversphere–atmosphere (LCA) coupling paradigm (Wu et al., 2009, 2012) and lithosphere–coversphere–atmosphere-ionosphere (LCAI) coupling paradigm (Qin et al., 2013). However, due to lack of AEF observations before and during the two major EQs, the abnormal changes in atmospheric parameters, such as aerosols and humidity, cannot be well linked to the changes in parameters of ground surface, such as microwave brightness temperature (MBT), thus the coupling process between the coversphere and atmosphere was not presented perfectly. Fortunately, in the seismogenic zone of the Luding EQ in 2022, the potential AEF disturbances before the EQ were recorded at four stations, which provided an excellent chance to study the abnormal features of AEF aroused by an EQ. In this study, the characteristics of pre-seismic AEF vibrations were analyzed, and the relationships between the AEF anomalies and the Luding EQ were carefully identified using multi-source auxiliary data. The mechanism of the seismic AEF anomalies was discussed by analyzing surface MBT variations and three-dimensional crustal stress distribution. Ultimately, four out of the nine potential AEF anomalies were determined to be earthquake-related.

## 2 Study area and data sources

### 2.1 Study area

The Ms6.8 Luding EQ, happened in Luding County, Sichuan Province, China, at 12:52 on 5 September 2022 (Beijing Time), with its epicenter located at 29.59°N, 102.09°E and a hypocenter depth of 14.5 km (Yang et al., 2022). The EQ occurred near the southeast Moxi section of the Xianshuihe fault (XSHF), which is a left-slip fault between the Bayan Har Block and the Sichuan-Yunnan Block (Ji et al., 2020). The study area was selected as [99°~106° E, 28°~32° N] in consideration of the Dobrovolsky formula (Dobrovolsky et al., 1979) and the geographical locations of the AEF observatories, in which there are Longmenshan faults (LMSF), the Anninghe fault (ANHF), the Longquanshan fault (LQSF) and the XSHF developed. Among these faults, the LMSF was the source of two significant earthquakes in the last few decades: the Lushan EQ in 2013 and the Wenchuan EQ in 2008, the latter having a catastrophic impact on lives. The AEF data used in this study were from four observatories called GAR, GUZ, SWG and LES. The first three locate nearby the XSHF while LES locates east to the southwest section of the LQSF. Figure 1 shows a complete overview of the study area.

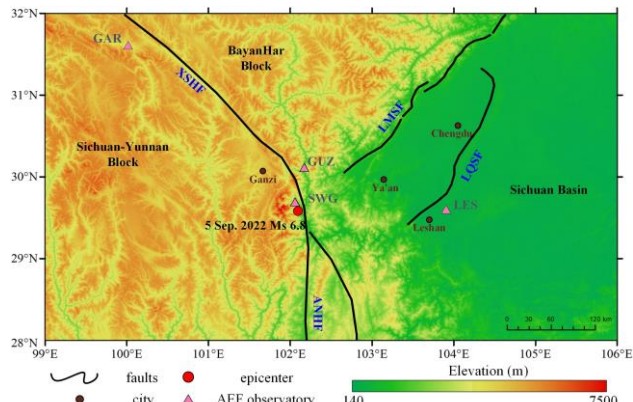

**Figure 1.** Distribution of the AEF observatories and topography in the study area. Background image is the digital elevation derived from Shuttle Radar Topography Mission (SRTM) datasets.

### 2.2  Data sources

### 2.2.1 Atmospheric electric field observatories

The GAR, GUZ and SWG were deployed by National Space Science Centre of Chinese Academy of Sciences (CAS) with instrument EFM-100 (Li et al., 2022). This type of instrument is independently developed by CAS, with a range of ±50 kV m-1, a relative accuracy of ±1 % and a resolution of 10 V m-1. The LES, which has a range of ±21.2 kV m-1, a relative accuracy of ±1 %, and a resolution of 3 V m-1, was deployed by China University of Geosciences (Wuhan) with instrument CS110 (Chen et al., 2021). The specific information of these AEF observatories is shown in Table 1. The GAR is located in a highland area in the southeast of Ganzi County, at an altitude of 3356 m above sea level (masl). The GUZ is located in Guzan Township, Kangding City, at an altitude of 1421 masl in the saddle, with the Dadu River flowing through it on the east side. The SWG is located in a valley in Yanzigou Town, Luding County, at an altitude of 2125 masl. The LES is located in Leshan City, with an altitude of 401 masl in the plain area, with flat terrain around it.

The intensity of AEF is measured according to the principle that a conductor can generate an induced-charge in an electric field. If a metallic conductor with surface area $S_c$ is exposed to an electrostatic field of electric field strength $E$, the charge density $\phi$ of the induced-charge generated on its surface can be expressed as:

$$\phi = \varepsilon_0 KE \tag{1}$$

where $\varepsilon_0$ represents the air dielectric constant and $K$ is the electric field distortion coefficient. The induced-

charge $Q$ can be expressed as:

$$Q = \phi S_C = \varepsilon_0 K E S_C \tag{2}$$

$$E = \frac{Q}{\varepsilon_0 K S_C} \tag{3}$$

**Table 1.** Key information of the AEF observatories.

| Site name | Longitude | Latitude | Altitude | Distance from the epicenter | Sampling frequency | Unit |
|-----------|-----------|----------|----------|-----------------------------|--------------------|------|
| GAR | 100.02° E | 31.61° N | 3356 masl | 298.97 km | | |
| GUZ | 102.17° E | 30.12° N | 1421 masl | 59.29 km | | |
| SWG | 102.07° E | 29.69° N | 2125 masl | 11.20 km | 1 s | Kv m-1 |
| LES | 103.91° E | 29.60° N | 401 masl | 175.67 km | | |

Consequently, by measuring the induced-charge amount, the strength of the AEF can be determined. An electric current will be generated when a metallic conductor is connected to the ground. It is known from electrical knowledge that if the conductor generates a continuously varying induced-charge, the measured intensity of the induced-current can be expressed as:

$$I = \frac{dQ}{dt} \tag{4}$$

The AEF meter sensor uses a moving piece and a stator to produce a continuously changing induced charge. As the moving piece begins to rotate, the stator is periodically exposed to the electric field or shielded under the moving piece and the two-stage circuit will receive a current signal of equal magnitude and opposite direction (Ji, 2022). Therefore, the $E$ can be deduced by measuring the intensity of the induced-current:

$$E = \frac{Q}{\varepsilon_0 K S_C} = \frac{\int I dt}{\varepsilon_0 K S_C} \tag{5}$$

### 2.2.2 Meteorological data and MBT

The AEF is influenced by a range of factors, including, but not limited to, meteorological conditions like clouds, rain, snow, and lightning, as well as global space weather activity such as solar activity and geomagnetic disturbances (Sun,1987). To accurately determine if the anomalous signals are caused by an EQ, it is essential to eliminate all potential influencing factors. In this research, the meteorological data used include low cloud cover (LCC), cloud base height (CBH), total precipitation (TP), and wind field, which is from the ERA5 reanalysis dataset provided by the European Centre for Medium-Range Weather Forecasts. The dataset is a globally complete and consistent data set formed by combining model data with global observation data using the laws of physics, which has been widely used for climatological studies. The LCC in the grid refers to the proportion of clouds that are below 2 km, CBH refers to the height of the lowest cloud base above the Earth's surface, TP is the cumulative value of liquid and frozen water falling on the Earth's surface over a period of time, and wind field includes the wind speed (WS) and direction at a height of 10m above the ground (Hersbach et al., 2023). The space weather data used include the geomagnetic index Dst from the World Geomagnetic Centre (WGC), the AE index from the Space Environment Prediction Center (SEPC) (Luo et al., 2013), and the sunspot numbers (SSN) from ESA.

The MBT data and surface soil moisture (SSM) data were also used to exclude local drought factors, as well as to analyze the potential accumulation of positive charges for the generation of AEF anomalies. The MBT data used is from the high-performance microwave radiometer AMSR-2 on board GCOM-W1, which is available at five microwave frequencies in both horizontal and vertical polarization (Imaoka et al., 2012). The UTC of MBT data has been converted to local time based on the satellite transit time. SSM data is derived from the GLDAS data set, which

represents a measure of moisture in the soil at a depth of 0~10 cm below the ground surface (Rodell et al., 2004).
Details of all the data used are shown in Table 2.

**Table 2.** Multi-source data for anomaly discrimination.

| Dataset | Data source | Temporal resolution | Spatial resolution | Unit |
|---|---|---|---|---|
| Low cloud cover | | | | / |
| Cloud base height | ERA5 | 1 h | 0.25° ×0.25° | km |
| Total Precipitation | | | | mm |
| 10m/100m Wind speed | | | | m/s |
| Dst | WDC | 1 h | / | nT |
| AE | SEPC | 10 mins | / | nT |
| Sunspot number | ESA | 1 d | / | / |
| Microwave brightness temperature | AMSR-2 | 1 d | 50 km×50 km | K |
| Surface soil moisture | GLDA V.2.1 | 3 h | 0.25°×0.25° | kg/m2 |

## 3 Results and analysis

### 3.1 Characteristics of local fair-weather AEF

To ascertain the periodic variations of AEF in the observatory, characterizing the background of FW-AEF is of
great importance. Consequently, obtaining a typical AEF curve as the FW-AEF background of GEC is crucial for the
identification and extraction of AEF anomalies. At present, the screening criteria for FW-AEF (Israelsson et al., 2001;
Harrison et al., 2018) cannot be fully standardized and need to be modified in conjunction with the local topographical
features, meteorological disturbances and geographical environment around the site. In this study, the following
screening criteria were set for obtaining FW-AEF: 1) no daytime rainfall, 2) low cloud cover closing to zero, 3) no
thunderstorms, 4) wind speed less than 8 m s-1 at 10 m above the ground, and 5) no long period of negative AEF
anomalies (to exclude anthropogenic influence and other uncertain factors). The sunrise occurs between 06:41 and
06:50 while the sunset taking place between 19:24 and 19:30 in the study area. The AEF data analyzed were from 1
May, 2022 to 30 September, 2022 for GAR, GUZ, SWG and from 1 August 2022 to 30 September 2022 for LES,
based on the data availability. After filtering and processing the AEF data, the daily variation curves of FW-AEF for
the four sites were obtained (Figure 2).
Figure 2 shows the 5-minute mean curve (left) of FW-AEF and satellite image from Google Earth (right). The AEF
zero value line is able to better identify negative AEF anomalies. Overall, the FW-AEF curve of GAR is characterized
by a single peak and two valleys, which displayed a shallow valley of 0.023 kV m-1 around 06:30, and then showed
a quick rise and reached the peak with 0.23 kV m-1 between 07:00 and 08:00, following by a gradual decline. The
second valley appeared between 19:00 and 20:00 with a valley of 0.015 kV m-1 at GAR. The FW-AEF at LES varied
more gently and behaved a single-peaked pattern, which showed a peak of 0.012 kV m-1 at around 14:00. The AEF
values of SWG and GUZ changed slightly before 12:00, but increased gradually to a peak about 0.14 kV m-1 for
SWG and 2.7 kV m-1 for GUZ around 19:10. The FW-AEF of SWG and GUZ were both single-peaked. The peak of
the FW-AEF curve of GUZ is much higher, which may be attributed to the particular topography of river valley and
the greater impact of human activity in the town.

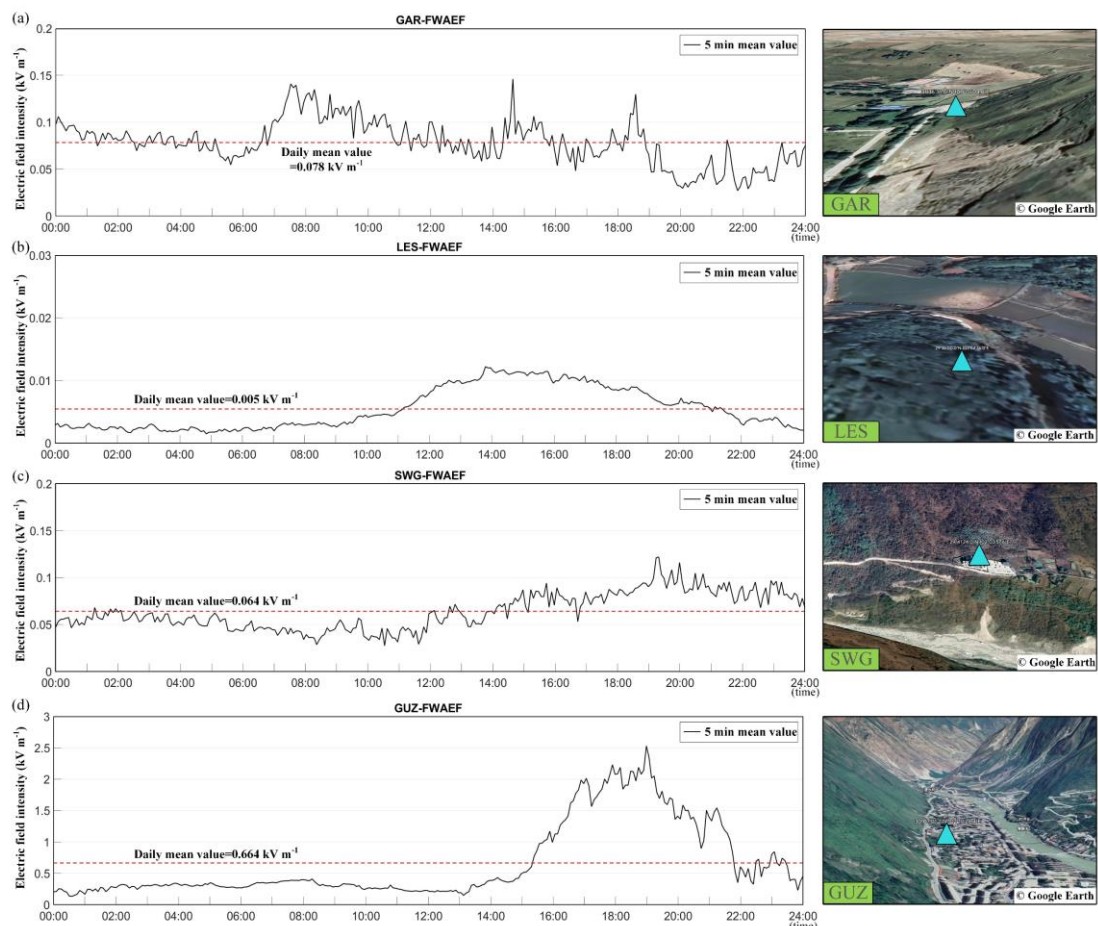

**Figure 2.** Daily variation of FW-AEF and Google Earth images at four sites of GAR (a), LES (b), SWG (c), and GUZ (d).

### 3.2 Identification of potential seismic AEF anomalies

Lightning, haze, meteorological events such as clouds and rain, and space weather events such as magnetic storms and solar activity are able to lead to changes in AEF. Global space weather events such as geomagnetic disturbances (Kleimenova et al., 2008), magnetic sub-bursts in the polar regions (Davis et al., 1966; Rastogi, 2005) and solar activity (Tacza et al., 2018) can also affect the AEF. The Dst, AE and SSN were utilized to represent the intensity of geomagnetic activity, polar magnetic storm, and solar activity, respectively. Figure 3(a) shows the variations of the three indices during 22 August to 5 September. According to the international practice, -50 nT < Dst < -30 nT represents weak magnetic storm activity (Loewe et al., 1997), AE < 100 nT represents the calm activity of the polar area magnetic substorms (Li et al., 2010), while 40 < SSN < 80 represents moderate solar activity. From 22 August to 25 August, the intensity of three activities was very weak, and AEF was not affected by space weather events during this period. Except for the period from 18:00 on 4 September to 06:00 on 5 September, all other time periods have Dst greater than -50 nT, which represents that the magnetic storm activity in the low-latitude region is weaker in these time periods, and the effect of this type of activity on the AEF can be ignored. Since the AE showed a higher value after 27 August, the AEF anomalies in the four hours before the EQ did not match with the high AE value in time. Therefore, even if AE fluctuated, the effect on AEF near the epicenter was very limited, being unable to cause the AEF anomalies.

In order to eliminate the effects of meteorology and space activities on the AEF, this study performed a time-series analysis of various climatological data (such as CBH, LCC, and TP) on an hourly or daily average basis. For the period of negative AEF anomalies, climatological data was examined to determine if non-seismic factors (meteorological parameters) occurred simultaneously, as depicted in Figure 3(b-j). The daily variation curves of AEF

from 22 August to 5 September and the hourly results of CBH, LCC and TP for the corresponding periods were retrieved, and nine periods of negative AEF anomalies with possible seismic activity factors were screened out from all the four sites. Specifically, there were four anomalies at GAR, four anomalies at LES, and one anomaly at SWG.

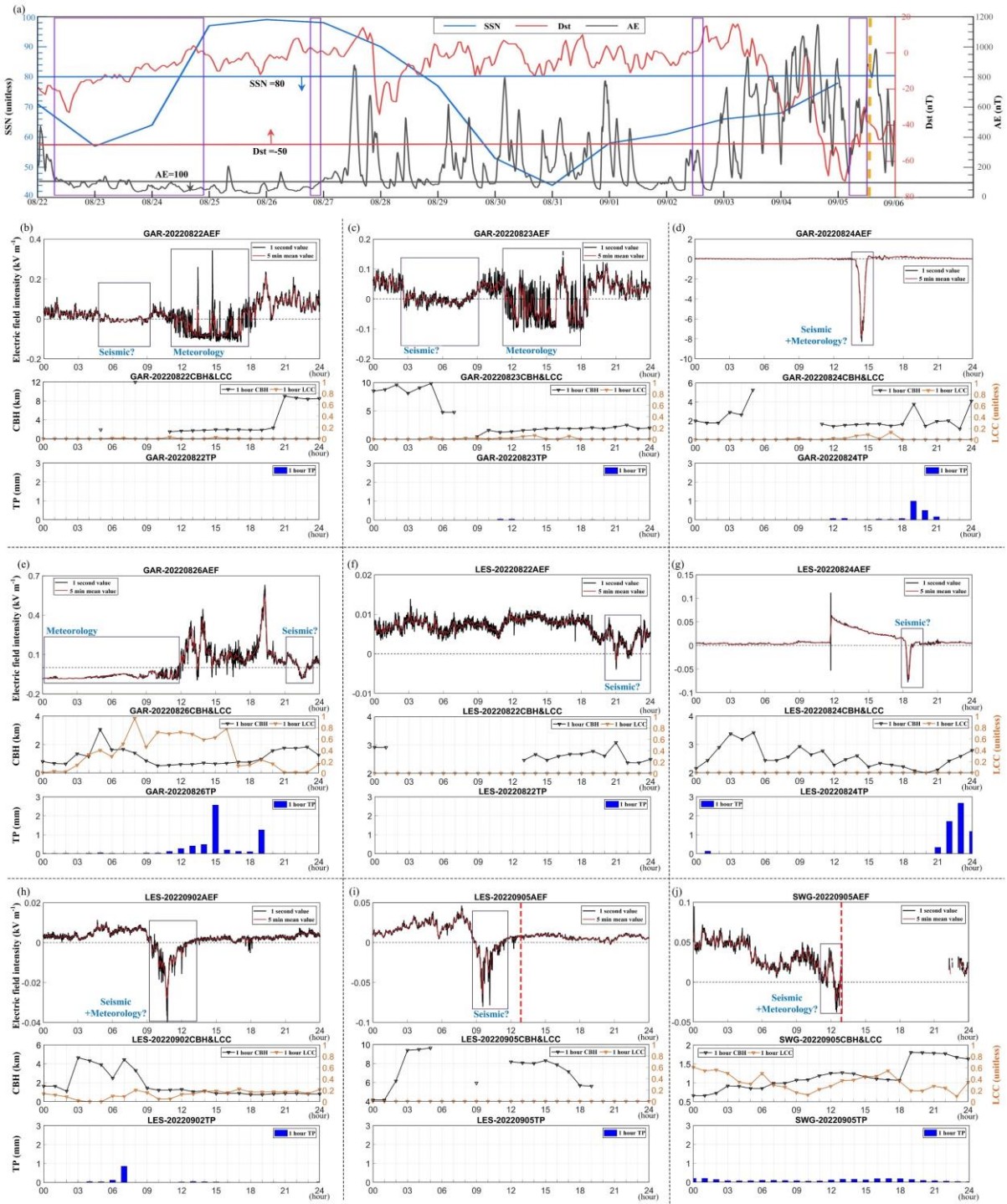

**Figure.3** Changes in SSN (blue), Dst (red) and AE (grey) from 22 August to 5 September. Horizontal thick lines of different colors indicate the thresholds of geomagnetic and solar activity for quiet periods represented by the corresponding indices, where the direction of the arrows represents a weakening of activity intensity (a). Nine negative AEF anomalies possibly related to the Luding EQ and hour-by-hour meteorological parameters (including CBH, LCC, TP) for the corresponding time periods (b-j).

For GAR, the AEF curve on 22 August and 23 August showed relatively similar patterns, with both negative AEF anomalies occurring twice during daytime. The first AEF anomaly appeared before 9:00, without low cloud and

precipitation, which indicates that it likely had been influenced by seismic activity. The second negative anomaly appeared between 12:00 and 18:00, with a small amount of precipitation at the beginning accompanied with a sudden drop in the height of cloud base and a rise in the amount of low cloud, which might had been caused by the combination of clouds and precipitation. The AEF anomaly of larger amplitude appeared between 13:00 and 15:00 on 24 August, with almost no precipitation, CBH less than 1.5 km and LCC less than 0.1. However, the influences of TP and cloud cover were much pronounced at this time, but the AEF did not show any negative anomaly. Hence, a mixture of meteorological and seismic activity was considered as a possible cause of the negative anomaly on 24 August. Two negative AEF anomalies appeared on 26 August, from 00:00 to 12:00 and from 21:00 to 23:00. In the period of first AEF anomalies, there was a prolonged small amount of TP, with a gradual rise in LCC, and a sudden increase in TP after the negative anomaly disappeared, which means that the AEF anomaly was probably resulted from the persistent precipitation washing away positive ions above the ground. The second segment of the AEF showed a decreasing trend at 21:00 and reached a minimal value at 22:30, returning to the FW-AEF level half an hour later, during which the LCC was close to zero and there was no TP, which was basically in line with the FW-AEF conditions. Hence, the second AEF anomalies on 26 August could be attributed to seismic activity.

For LES, the AEF anomaly appeared on 22 August between 20:00 and 23:00, with no precipitation and no low clouds existed, and the CBH was greater than 2 km throughout the day, which fully met the criteria for the FW-AEF. zero precipitation and no low clouds existed during the period of AEF anomaly occurring between 18:00 and 19:00 on 24 August. The negative anomalies appeared on 2 September and 5 September both appeared between 08:00 and 12:00, with LCC less than 0.1 during the anomalies, but with high precipitation before and slightly higher precipitation after the anomalies. There was no TP and LCC on 5 September, with the CBH greater than 4 km all day. Therefore, it can be determined that the negative AEF anomalies appeared on 22 August and 24 August, and 5 September might had been influenced by seismic activity, while the negative AEF anomaly appeared on 2 September could be attributed to the mixture of meteorological and seismic activity.

**Table 3.** Details of each parameter of the anomalous AEF time periods.

| Site | Time period of AEF anomalies | CBH (km) | LCC (unitless) | TP (mm) |
| --- | --- | --- | --- | --- |
| GAR | 8/22 05:00-09:00 | >10 | 0 | 0 |
| GAR | 8/23 03:00-08:00 | >5 | 0 | 0 |
| GAR | 8/24 13:00-15:00 | >1.5 | <0.1 | <0.01 |
| GAR | 8/26 21:00-23:00 | >1.8 | <0.01 | 0 |
| LES | 8/22 20:00-23:00 | >2 | 0 | 0 |
| LES | 8/24 18:00-19:00 | >2 | 0 | <0.01 |
| LES | 9/02 09:00-12:00 | >1 | <0.1 | <0.01 |
| LES | 9/05 08:00-11:00 | >6 | 0 | 0 |
| SWG | 9/05 12:00-13:00 | >1 | <0.2 | <0.1 |

For SWG, the AEF on 5 September showed a downward trend from 04:00, dropping to a negative level at around 12:15, and the negative state lasted for 35 minutes until 12:50, reaching a minimal value of -0.04 kV m-1 at 12:29. Due to the proximity of the site to the epicenter, the EQ triggered a power outage in the adjacent area, resulting in a data missing at SWG after 12:53. The site had light rain all day, with precipitation less than 0.2 mm, CBH greater than 500 m and LCC less than 0.6. However, as compared to the FW-AEF at the SWG, it can be found that the decreasing trend of AEF from 04:00 to 09:00 coincided perfectly with the simultaneous FW-AEF changes, and there was no significant change in magnitude, so the effect of meteorological activity on AEF on 5 September was not particularly significant. In summary, the negative AEF anomaly appeared between 12:00 and 13:00 could be

attributed to a combination of meteorological and seismic activity. Details of each parameter related with the AEF
negative anomalies are shown in Table 3.

## 4 Verification and Scrutinization

The fluctuation of AEF, which is influenced by global thunderstorm activity, is primarily dependent on the
concentration of near-surface atmospheric ions. Atmospheric ions exist in the air even in fair-weather conditions, and
contribute to the atmosphere's electrical conductivity. The concentration of atmospheric ions can be directly or
indirectly altered by factors such as rainfall, low clouds, haze, and aerosols. Therefore, it is important to understand
why the near-surface ion concentrations changed prior to the earthquake in order to uncover the underlying correlation
between pre-seismic AEF anomalies and the Luding earthquake.

### 4.1 P-holes manifestation verified by MBT, SSM and Geology

Some researchers explained well the reasons for positive MBT anomalies preceding EQs from the perspective of
P-hole theory (Qi et al., 2022; Ding et al., 2022), and the AEF anomaly was mentioned in the conceptual diagrams of
LCAI coupling process. Mao et al. (2020) demonstrated that the microwave dielectric constant decreases on rock
surfaces under compressive loading experimentally. The pre-seismic MBT anomalies in the Qinghai-Tibet Plateau
region have been extensively discussed in previous studies. Liu et al. (2023) analyzed the relationship between MBT
anomalies and extensional faults. Qi et al. (2021a) discovered the positive MBT anomaly preceding the May 2008
Wenchuan EQ, and explained the geological influence on the positive MBT anomaly based on P-hole theory. When
P-holes are transferred to the surface, it not only changes the dielectric constant, but also causes air ionization near
the surface. According to the researches on seismic MBT anomalies in the same area of this study, MBT at the low
frequency with horizontal polarization performed better (Qi et al., 2021a, 2023).
Therefore, MBT data at 10.65 GHz with H polarization was used in this study. To analyze the potential surface
microwave dielectric changes caused by the seismicity, the MBT anomalies during 15d before the Luding EQ were
obtained by using STW-TSM (Qi et al., 2020). Theoretically, MBT depends largely on the surface emissivity, which
lies on the dielectric constant and the physical temperature (Ulaby et al.,1981). The surface dielectric constant will
increase and results in a decrease in MBT when SSM rises. Temperature changes, precipitation processes, and the
rise and fall of the underground water level all lead to changes in SSM, which can also affect surface MBT. In order
to identify seismic MBT anomalies, it is necessary to use SSM data to discriminate the potential MBT anomalies. In
this research, SSM residuals from the surface to 10 cm underground were obtained by subtracting the average value
of the same time period of the background year from the seismic year data, which was used to discriminate the local
drought factor.
Figure 4 shows the residual MBT and residual SSM images from 22 August to 5 September, 2022. Overall, the
positive MBT anomalies appearing from 22 August to 1 September were mainly concentrated in the plains to the east
of the LMSF, the mountainous areas to west and northwest of the XSHF (mainly bare land), and the southeast corner
of the Bayan Har Block. Positive MBT anomalies gradually appeared in various areas on 22 August, with its range
expanded to the maximum and its amplitude reaching to 10~15 K on 25 August. The positive MBT anomalies still
existed in few areas after 28 August, and generally dissipated after 2 September. The residual SSM remained a lower
value in most of the regions from 22 August to 28 August, and there was also a significant increase on 30 August and
a slow decline on the coseismic day. Therefore, the positive MBT anomalies due to local drought factors (SSM drop)
should be excluded hereinafter. Specifically, the areas of positive MBT anomalies were distinguished by dashed
polygons with different colors in Figure 4. The sequential positive MBT anomalies were zoned as zone I~VII, their
spatial relations to the surface lithology were shown in Figure 5.

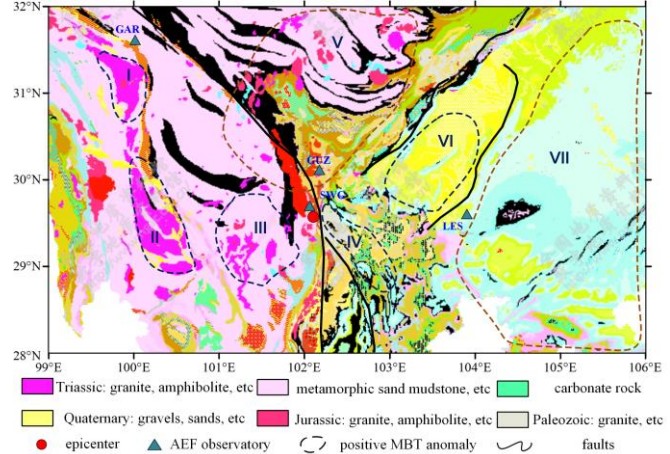


**Figure 4.** Residual MBT images at 10.65 GHz with H polarization and residual SSM from 22 August to 5 September. Blue polygons
represent relatively low or no change in SSM while red polygons represent a significant drop in SSM.

**Figure 5.** Distribution of surface lithology in the study area (image from the National Geological Archive).
Zone I was in the northwest corner of the study area, near Ganzi section of the XSHF. The MBT anomaly in zone
I appeared on 22 August with an amplitude of 8 K (14d before the EQ), followed by a gradual decrease until 28
August. It appeared again on 31 August, with the positive anomaly spreading southward on 1 September and
dissipating after 2 September (3d before the EQ). SSM residuals in zone I decreased by a lower amount than the
surrounding area from 22 August to 28 August, and remained almost unchanging after 31 August. The residual MBT
and SSM did not conform in time to the physical process that SSM decrease leads to the rise of MBT. Likewise, the
spread of the positive MBT anomaly to the north on 28 August and the persistence of the MBT anomaly on 30 August
in zone II cannot be well explained by SSM change either.
The MBT anomalies in zone III were generally striped along XSHF, which started to appear on 23 August (13d
before the EQ), with its amplitude increasing on 24 August, basically dissipating on 26 August. The MBT anomalies
appeared again on 28 August with the maximum amplitude of about 10 K, then gradually weakened before the EQ.
The residual SSM in zone III was low on 23 August and 24 August, which was inconsistent with the amplitude
increase in MBT anomaly. A small decrease in SSM residuals appeared in zone III on 28 August, which was consistent
with the appearance of the positive MBT anomaly. There was a good spatio-temporal correlation between the positive
MBT anomaly and SSM decline.
Positive MBT anomalies in zone IV gradually became apparent on 24 August, more pronounced in the north on 26
August and in the south on 28 August. The SSM residuals were in a state of low negative value from 24 August to 28
August and no significant change was detected over time. This was also the case for zone VI, where the variation in
SSM was slight during the MBT anomalies from 24 August to 28 August. The positive MBT anomalies in zone V
mainly appeared on 24 August and 25 August, with a large range of high amplitude. The SSM in zone V decreased
during the same period, and then SSM residuals gradually increased, which corresponded well with the positive MBT
anomalies on the space and time scales. The same situation also happened in zone VII as for zone V from 23 August
to 31 August.
After analyzing the spatio-temporal evolution of MBT residuals and SSM residuals in the seven zones of MBT
anomalies, the appearance of positive MBT anomalies in five zones (I, II, III, IV and VI) were thought to be related
to the Luding EQ. Accordingly, the positive MBT anomalies associated with seismic activity in these five zones were
further analyzed by introducing the lithology distribution map and numerical simulations of the CSFA. Figure 5
shows the surface lithology in the study area. According to P-hole theory, the production and convergence of P-holes
occurs in rocks with peroxy-defects (peroxy-bonded) structures, and the main carriers of peroxy-bonded are low-
crystalline minerals including quartz and feldspar (Freund, 2002). As can be seen in Figure 5, the lithology of zone I,
II and III is dominated by granites, metamorphic sandstones and other rocks containing quartz and feldspar
components with peroxide defect structures. Zone VI is dominated by the Quaternary, the geological strata is
relatively loose and the major lithology is sand and gravel consisting of granular quartz, feldspar, mica, etc. Zone IV
has a more complex lithological distribution behaving fewer minerals with peroxy- defects structure than the others,
and the appearance of positive MBT anomalies in zone IV were shorter in duration and relatively small in area.
Therefore, it was considered that zones I, II, III and VI are more prone to positive MBT anomalies following P-hole
accumulation.
The uneven distribution of crustal stress and its gradually accumulation are the main causes of tectonic seismicity.
Based on the Crust 2.0 model and stratigraphic data (Shan et al., 2009; Li et al., 2022), a three-dimensional (3D)
stratigraphic model was constructed using the 3D finite element method to simulate the CSFA due to seismic tectonics
at a time scale of 1000 years. The stratigraphic model had an east-west width of 1000 km, a north-south width of 800
km and a depth of 83 km, and the simulated crustal stress within the study area of this research was intercepted
(Figure 6). Historical EQ catalogs from 1770 to 2022 were selected to make seismic validation of the simulated CSFA.

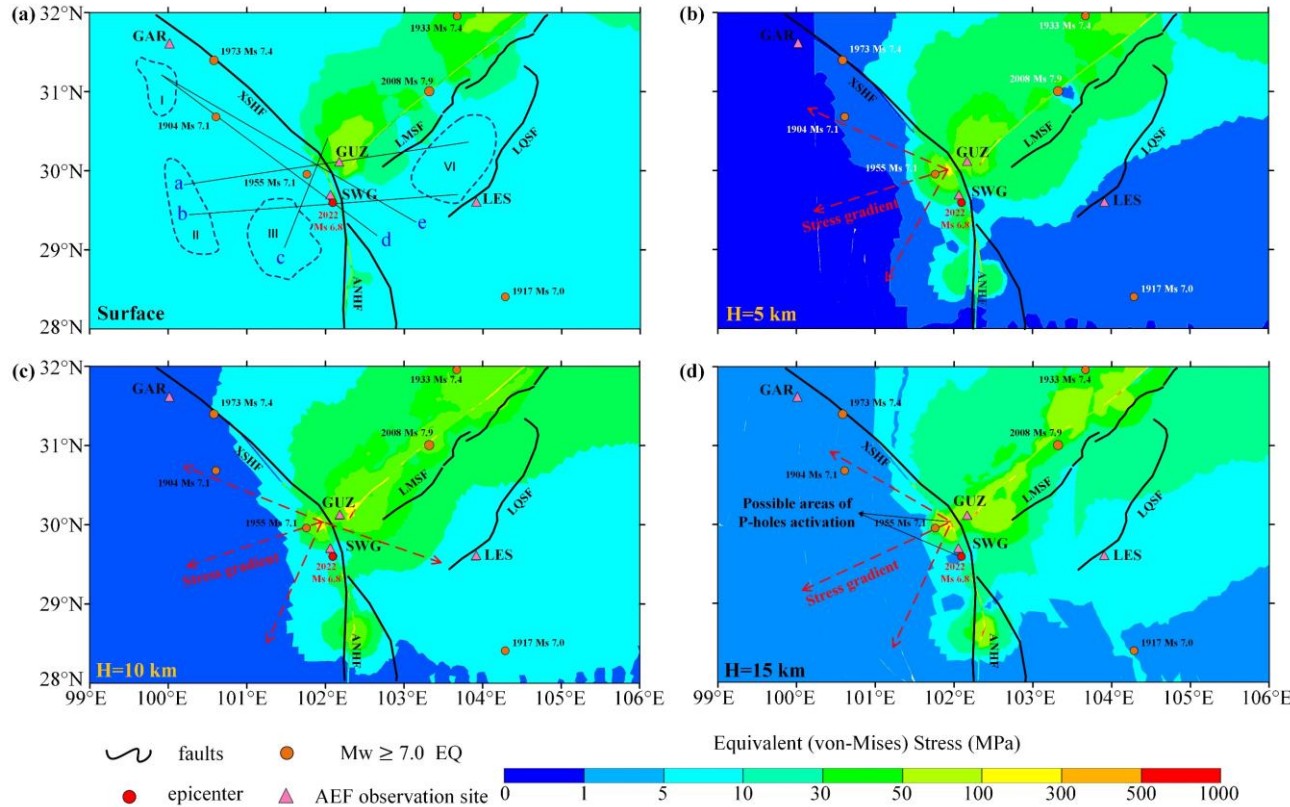


**Figure 6.** Spatial distribution of equivalent stress at ground surface (a), 5 km deep (b), 10 km deep (c) and 15 km deep (d).
As shown in Figure 6, the equivalent stress intercepted at different depths was used to reflect the crustal stress
background. In the map of the CSFA at the depth of 15 km, crustal stress was mainly concentrated in three places,
i.e., the left side of the southeast section of XSHF, the area along LMSF and the right side of ANHF. Large EQs
(magnitude of 7 or higher) had occurred in all the three places in history. The activated P-holes are to flow from the
seismic source area to upper crust in response to the direction of maximum stress gradient (Freund et al., 2006, 2021;
St-Laurent et al., 2006). Compared with the stress concentration areas at the depth of 15 km, the size of the surface
stress concentration areas as well as the stress magnitude are weakened, indicating that there was an overall upward
stress gradient between the 15 km depth plane and the ground surface.
In addition, the high stress areas are mainly concentrated in the central study area as well as in the northeast, with
lower stress appearing in the southeast and southwest, indicating the existence of a stress gradient toward the
southeast and west sides. P-holes generation would occur not only at the hypocenter, but also in areas of high stress
concentration. Based on the simulated CSFA, the hypocenter and its nearby high stress area were selected as the
places where the P-holes activations were generated (at a depth of about 15 km). The stress gradients from the
hypocenter or the nearby high stress areas to the four seismic MBT zones were calculated by dividing the stress
difference by the distance. The results are shown in Table 4, and the corresponding stress profiles are described in
Figure 7. P-holes could be activated from the hypocenter or from its nearby high stress areas, thus there was the
possibility of P-holes transferring along the stress gradient to all the four seismic MBT regions. It is also clear that
the closer to the hypocenter or the high stress area, the higher the stress gradient.

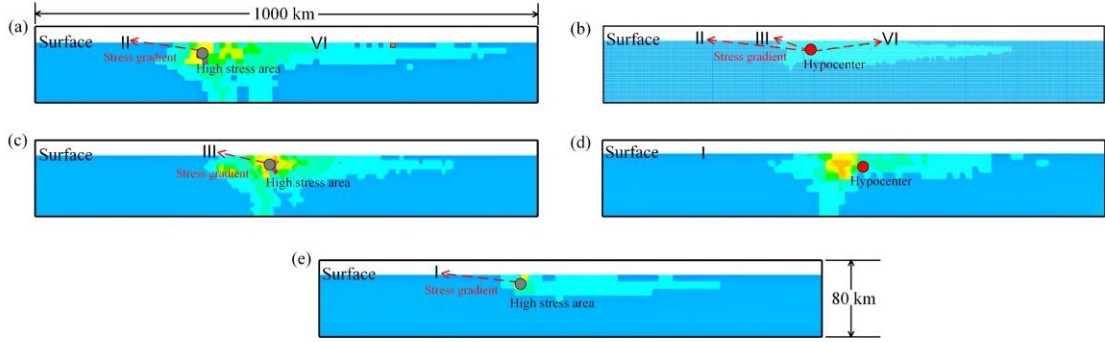


**Figure 7.** The vertical sections of crustal stress from the hypocenter or nearby high stress area to the four seismic MBT zones. Subplot
(b) and (d) are vertical profiles through the hypocenter, while subplot (a), (c) and (e) are the vertical profile started from the nearby high
stress concentration areas.
**Table 4.** Stress gradients from two P-hole activation areas to the four residual MBT regions in Figure 4.

| Zones | Stress magnitude (MPa) | Distance to epicenter (km) | Distance to high stress area (km) | Stress gradient from hypocenter | Stress gradient from high stress area |
|-------|-------|-------|-------|-------|-------|
| I | 18.72 | 259.09 | 219.53 | 0.41 | 76.45 |
| II | 14.73 | 190.46 | 183.22 | 0.58 | 91.62 |
| III | 18.50 | 88.57 | 109.88 | 1.20 | 152.73 |
| VI | 58.74 | 162.18 | 181.92 | 0.41 | 92.02 |

In summary, the positive MBT anomalies that appeared in zones I, II, III, and VI during 14d before the EQ were
identified to be possibly related to seismic activity. Positive MBT anomalies in ground surface due to CSFA indicates
the occurrence of P-hole aggregation, which provides the conditions for air ionization to exist in the near-surface
atmosphere.
**4.2 Scrutinization of seismic AEF anomalies**
After screening for negative AEF anomalies, it was found that there were noticeable differences in both space and
time between the sites and the regions with positive MBT anomalies. Therefore, it should be considered that positive
ions generated by P-hole ionized air can spread in the atmosphere and drift to the sites with the wind field. By
analyzing the near-surface wind direction and wind speed, it can be determined if there exists an appropriate wind
field between the site and the regions with positive MBT anomalies. In this study, wind field data at 10 m above the
ground with a temporal resolution of 1 hour were used. Considering that the wind speed in the entire study area was
below 8 m s-1, the distance over which atmospheric ions could be transported by the wind field was greatly limited.
This limitation was a result of neutralization caused by electrostatic interactions and the absorption of aerosols.
(Wright, et al., 2020). Therefore, only the wind field in the zone of positive MBT anomaly nearest to the AEF sites
were took into consideration.
The occurrence of negative AEF anomalies can be divided into two categories. The first one is that the wind field
in the MBT anomaly area did not show a trend moving towards the AEF site, such as the negative AEF anomaly at
GAR on 22 August, 23 August and 26 August, and LES on 24 August and 2 September. In Figure 8(a), the times of
MBT residual and SSM residual images were both 02:00 on 22 August, and the wind direction and speed were at
four moments of 01:00, 04:00, 07:00 and 09:00. The wind direction from zone I to GAR was not indicated before
and during the MBT anomaly, and the wind speed was too low to transfer the positive ions generated on the ground
surface in zone I to GAR, thus the negative AEF anomaly at GAR on that day was not resulted from seismic activity.
The other four days were all in the same situations as this day.

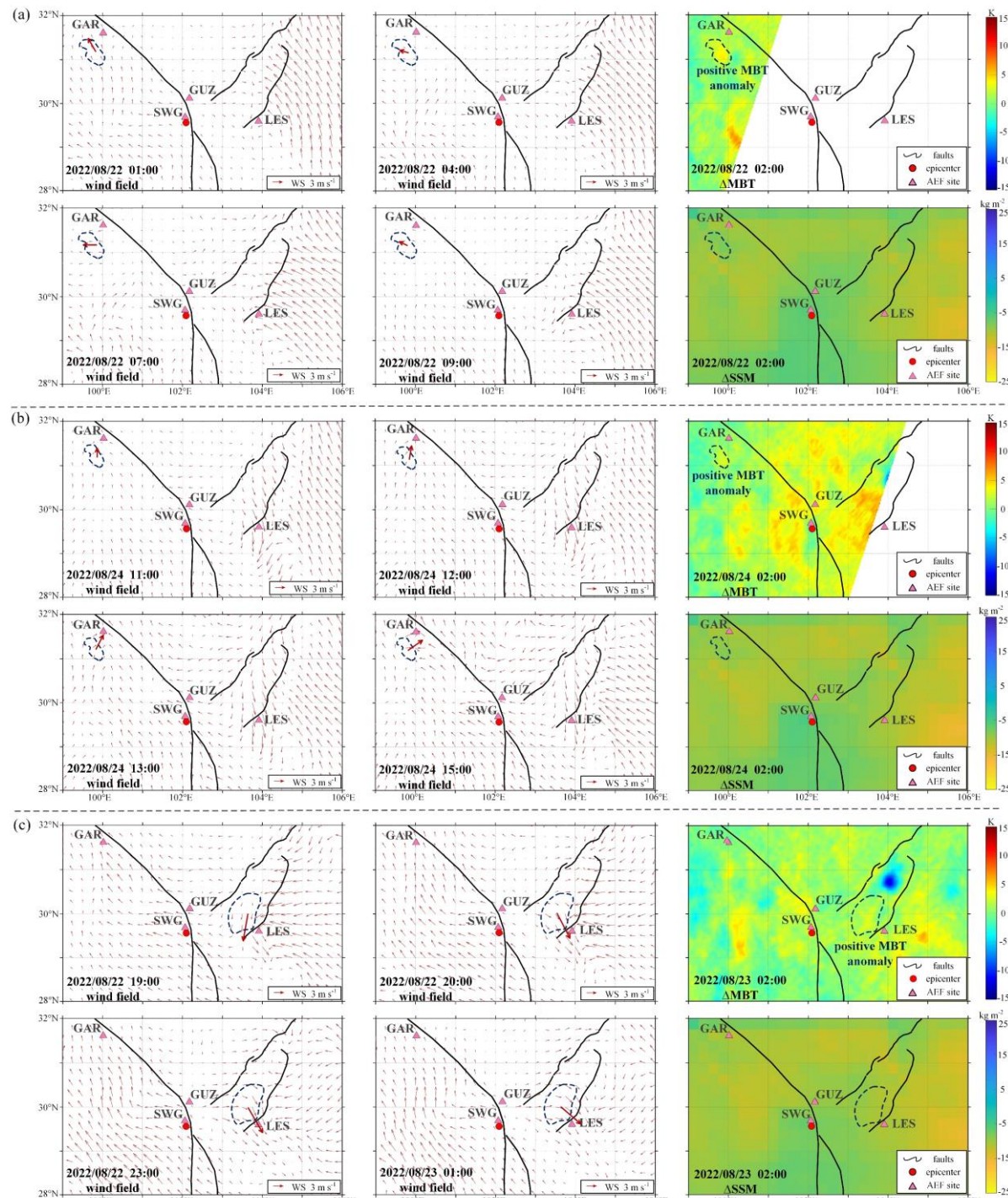

**Figure 8.** Wind field, MBT residuals and SSM residuals in the study area on 22 August (a) and 24 August (b) for GAR, and on 22 August for LES (c), 2022.

The second category is that the wind field in the area of positive MBT anomalies was pointed towards the AEF sites, such as the negative AEF anomalies at GAR on 24 August and LES on 22 August. In Figure 8(b), the AEF anomaly at GAR appeared between 13:00 and 15:00 on 24 August. Before 11:00, the wind direction in zone I was slightly to the west of the site, the wind direction began to deflect to the northeast at 12:00 with the direction pointing to the site and keeping the direction until 14:00, during which the wind speed gradually increased. The wind deflected to the northeast again at 15:00, and then gradually deviated. The wind field changes during this period coincided well with the appearance of the negative AEF anomaly, and the wind field at the site in the end of the anomaly period was

also changing, so a longer period of AEF anomaly caused by positive ions staying and accumulating at the site could be ruled out, which was also consistent with the AEF returning to positive levels after 15:00. Therefore, the negative AEF anomaly at GAR on 24 August likely had been influenced by seismic activity. In Figure 8(c), the AEF anomaly appeared between 20:00 and 23:00 on 22 August at LES. The wind field was pointing west of the AEF site before the anomaly appeared, and then veered east and pointed towards the site from 20:00 to 23:00. Then, the wind field continued to veer east and drifted away from the site at 01:00 the next morning. The changes in wind field corresponded well to the time of the appearance of the negative AEF anomaly, which shows that seismic activity might had impacted on the AEF anomaly at that time.

On the coseismic day, although there were AEF anomalies appeared at the LES and SWG, the MBT data were missing due to satellite coverage. Taking into account that the surface lithology at both sites was similar to that of the closest MBT anomaly area, and the negative AEF anomaly emerged only 4 hours prior to the EQ, it can be inferred that a localized P-hole aggregation phenomenon may have occurred in the immediate vicinity as a direct consequence of seismic activity. This phenomenon would have led to air ionization, thereby altering the vertical AEF.

In conclusion, the negative AEF anomalies observed at GAR from 13:00 to 15:00 on 24 August and at LES from 18:00 to 19:00 on 22 August were believed to be potentially related to the surface P-hole accumulation caused by seismic activity. The anomalous AEF signals at LES and SWG 4 hours before the EQ on September 5 were considered to be associated with localized changes in atmospheric ion concentrations due to seismic activity during the short imminent stage of the Luding EQ.

## 5 Conclusions

In this study, historical AEF data from four AEF observatories, namely GAR, LES, GUZ, and SWG, were collected to construct and analyze the FW-AEF. The curves of FW-AEF exhibited positive fluctuation states and were characterized by single or double peaks. Subsequently, the AEF variations occurring 15 days prior to the Luding EQ in 2022 were meticulously examined, using the FW-AEF as a reference state. As a result, nine AEF negative anomalies (four at GAR, four at LES, and one at SWG) were identified as potentially related to the Luding EQ, which was reached by analyzing meteorological parameters including CBH, LCC, TP, and space weather parameters including Dst, AE, and SSN. Furthermore, the MBT residuals during the 15 days prior to the Luding EQ were comprehensively analyzed in conjunction with SSM, geological maps, and numerically simulated CSFA, the geophysical environment for high-stress concentration in crust, positive charge carriers transfer and accumulation to Earth's surface were proved to exist, which meet the condition of producing seismic AEF anomalies. Furthermore, ground-based wind field data were utilized to investigate the causes of the negative AEF anomalies, taking into account the spatial differentiation between the AEF observatory locations and the areas where positive charge carriers accumulate. The confirmed causes of the AEF anomalies are listed in Table 5.

**Table 5.** Summary of the AEF anomalies before the Luding EQ on 5 September, 2022.

| AEF sites | Duration of AEF anomalies | Presence of meteorological effect | Presence of seismic effect | Causes of negative AEF anomalies |
|---|---|---|---|---|
| GAR | 8/24 13:00-15:00 | √ | √ | seismic and meteorological effects |
| GAR | 8/26 21:00-23:00 | √ | × | meteorological effect |
| LES | 8/22 20:00-23:00 | × | √ | seismic effect |
| LES | 9/02 09:00-12:00 | √ | × | meteorological effect |
| LES | 9/05 08:00-11:00 | × | √ | seismic effect |
| SWG | 9/05 12:00-13:00 | √ | √ | seismic and meteorological effects |

The negative seismic AEF anomalies appeared preceding the Luding EQ in 2022 were ascribed to the positive

charge carriers generated in areas with high stress concentration and accumulated on the ground surface. These charge carriers were capable of ionizing the near-surface air in the surrounding atmosphere, leading to the observed anomalies. This action mechanism serves as a link to establishing the coupling process between the coversphere and the atmosphere, which is crucial for understanding multiple seismic anomalies. The work carried out in identifying and assessing seismic AEF anomalies as reported in this study is anticipated to offer a valuable example for future research in this field.

**Data availability**

All data can be provided by the corresponding authors upon request.

**Author contributions**

LW, XW and YQ designed the framework of the manuscript; XW and YQ wrote the manuscript draft; LW and YQ polished the manuscript; JL and WM performed the mechanical simulation; XW and JL completed the visualization, LW provided the funding. All authors have read and agreed to the published version of the manuscript.

**Competing interests**

The authors declare that they have no conflict of interest.

**Financial support**

This work was supported by the Key Program of National Nature Science Foundation of China (41930108).

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
