# Peer review of "Characteristics and mechanisms of near-surface atmospheric electric field"

_EGUsphere, 2023_

## Community Comment (CC3)

Thank you very much for your valuable comments on our manuscript, and our responses are listed below.

**Question 1:**
Figure 2. It is not clear if the figure represents a typical day of FW-AEF or a daily mean curve.

**Reply:**
The FW-AEF curve in Fig. 2 represents the AEF curve that can reflect the local natural state (i.e., fair weather), which is calculated from a large number of filtered historical data. The FW-AEF curve exhibits the characteristics of a cycle in cyclical change (i.e., 24 hours), instead of a typical day of FW-AEF or a daily mean curve. For the AEF sites, the frequency of actual data acquisition is relatively high, and we have made smooth operations (5-min-mean FW-AEF). In the main text, we have stated the period of the data used from each site in the process of screening the fair-weather conditions, and the AEF data that conform to FW-AEF conditions were selected to be summed and then averaged to get the diurnal variation of FW-AEF curve of the site.

**Question 2:**
Lines 172-175. Among magnetic indices, only Dst is used. However, some penetrating field currents can appear in middle latitudes on occasion of some aurora activity. It would be more conservative to consider also AE index. When it is lower than 100, it is practice to neglect the presence of that kind of currents. Could you please check if AE is lower than 100 in the period of interest?

**Reply:**
Thank you for your suggestions. Here we added the changes of AE index from 0:05 on 22 August, 2023 to 23:55 on 5 September, 2023, with the temporal resolution of 10 mins, which are derived from the Space Environment Prediction Center (SEPC) (Luo et al., 2013). It is suggested that the ring current is energized occasionally by the injection of particles from even the sub-auroral latitudes along the earth's magnetic field lines (Rastogi, 2005), and the AE index can characterize the strength of magnetic sub-bursts in the polar regions (Davis et al., 1966). In Fig. 3(a), the SSN is still the value of the daily change, and the Dst was changed to the value of the hourly variation. AE < 100 nT represents the calm activity of the polar area magnetic substorms, while AE=100-300 nT represents modest activity of the polar area magnetic substorms. When AE > 300 nT, the magnetic storm activity in the polar region is intense (Li et al., 2010).

The AE index remained essentially <100 nT from 12:00 on 22 August to 24:00 on 27 August, so the polar region magnetic storm activity was weak during that period and its effect on the AEF can be ignored. The AE was higher in other periods, some of which even reached >1000 nT, such as on 4 September. Two of the four identified anomalies in the manuscript occurred in the quiet period of polar magnetic storm activity (8/22 20:00-23:00 at LES, 8/24 13:00-15:00 at GAR), while the other two appeared within 4 hours before the EQ and disappeared after the EQ. Since the AE showed a higher value after 27 August, the AEF anomalies in the four hours before the EQ did not match with the high AE value in time. Therefore, even if AE fluctuated, the effect on AEF near the epicenter was very limited, being unable to cause the AEF anomalies. All the information will be added in the revised version.

[Figure]

Figure.3(a) Changes in SSN (blue), Dst (red) and AE (grey) from 22 August to 5 September. Horizontal thick lines of different colors indicate the thresholds of geomagnetic and solar activity for quiet periods represented by the corresponding indices, where the direction of the arrows represents a weakening of activity intensity. The orange dashed line represents the moment of the EQ's onset, and the purple solid box represents the periods of the AEF anomalies.

AE data was also added to Table 2 (Multi-source data for anomaly discrimination).

**Table 2.** Multi-source data for anomaly discrimination.

| Data name | Data source | Temporal Resolution | Spatial Resolution | Unit |
|---|---|---|---|---|
| Low cloud cover | ERA5 | 1 h | 0.25°×0.25° | / |
| Cloud base height | | | | km |
| Total Precipitation | | | | mm |
| 10m/100m Wind speed | | | | m/s |
| Dst | WDC | 1 h | / | nT |
| AE | SEPC | 10 mins | / | nT |
| Sunspot number | ESA | 1 d | / | / |
| Microwave brightness temperature | AMSR-2 | 1 d | 50 km×50 km | K |
| Surface soil moisture | GLDA V.2.1 | 3 h | 0.25°×0.25° | kg/m2 |

Rastogi, R. G.: Magnetic storm effects in H and D components of the geomagnetic field at low and middle latitudes, J. Atmos. Sol.-Terr. Phys., 67, 665-675, https://DOI10.1016/j.jastp.2004.11.002, 2005.

Davis, T. N., and Sugiura, M.: Auroral electrojet activity index AE and its universal time variations, J. Geophys. Res., 71, 785-801, https://agupubs.onlinelibrary.wiley.com/doi/epdf/10.1029/JZ071i003p00785?saml_referrer, 1966.

Luo, B. X., Li, X. L., Temerin, M., and Liu, S. Q.: Prediction of the AU, AL, and AE indices using solar wind parameters, J. Geophys. Res.: Space Phys., 118, 7683-7694, https://agupubs.onlinelibrary.wiley.com/doi/full/10.1002/2013JA019188, 2013.

Li, W., Thorne, R. M., Bortnik, J., Nishimura, Y., Angelopoulos, V., Chen, L., McFadden, J. P., and Bonnell, J. W.: Global distributions of suprathermal electrons observed on THEMIS and potential mechanisms for access into the plasmasphere, J. Geophys. Res.: Space Phys., 155, A00J10, https://agupubs.onlinelibrary.wiley.com/doi/full/10.1029/2010JA015687, 2010.

**Question 3:**

It shows at its top panel the daily mean values of Dst and SSN. However, this value for Dst is not appropriate and cannot well characterize the level of magnetic activity. It would be more appropriate its hourly value.

**Reply:**

This issue was corrected accordingly in Fig.3(a) in response to Question 2 by converting Dst daily averages to hourly variations. The Dst > -50 nT and < -30 nT represents weak magnetic storm activity, Dst > -100 nT and < -50 nT represents moderate magnetic storm activity (Loewe et al., 1997). Compared with the daily mean Dst, the hourly values are more representative of the magnitude of the intensity of magnetic storm activity at low latitudes. Except for the period from 18:00 on 4 September to 06:00 on 5 September, all other time periods have Dst greater than -50 nT, which represents that the magnetic storm activity in the low-latitude region is weaker in these time periods, and the effect of this type of activity on the AEF can be ignored.

Loewe, C. A., and Prölss, G. W.: Classification and mean behavior of magnetic storms, J. Geophys. Res.: Space Phys., 102, 14209-14213, https://agupubs.onlinelibrary.wiley.com/doi/pdf/10.1029/96JA04020, 1997.

**Question 4:**

How did you define the 0 value of AEF?

**Reply:**

First of all, AEF is a vector, the AEF values measured at the site are the vector superposition of all the AEFs in the detection range. A value of zero does not mean that the electric field no longer exists, it represents that the electric field strength at a certain location at a certain time is zero. Secondly, the zero value may be caused by the presence of an additional electric field of equal magnitude but opposite direction, thus counteract the original AEF. The AEF zero value line in Fig.3 is able to better identify negative AEF anomalies. This information will be explained in the revised version.

---

## Community Comment (CC5)

Thank you very much for your valuable comments on our manuscript, and our responses are listed below.

**Question 1:**
On line 33, "atmospheric field" should be replaced by "atmospheric electric field";

**Reply:**
We have corrected the problem you mentioned in the manuscript. In accordance with the principle that the first occurrence is indicated by 'atmospheric electric field' and the rest of the text by 'AEF', the problem is also searched and corrected in the full text.

**Question 2:**
On line 33, "non-thunderstorm or sunny areas" is not correct. Even in the absence of thunderstorms but the presence of charged clouds or haze, the atmospheric electric field may be negative. This statement could be considered to be changed to "undisturbed fair areas".

**Reply:**
We have corrected the problem you mentioned in the manuscript.
Original text:'In the background of GEC, a direct current (DC) atmospheric field with an amplitude of around 130 V/m is always present in global non-thunderstorm or sunny areas (Sun,1987).'

Revised version: 'In the background of GEC, a direct current (DC) atmospheric electric field with an amplitude of around 130 V/m is always present in undisturbed fair areas (Sun,1987).'

**Question 3:**
On lines 101-103, the detailed differences between these two types of instruments should be elaborated in more detail.

**Reply:**
Original text:'The GAR, GUZ and SWG were deployed by National Space Science Centre of Chinese Academy of Sciences with instrument EMF-100 (Li, 2022), and the LES was deployed by China University of Geosciences (Wuhan) with instrument CS110 (Chen et al.,2021).'

Revised version: 'The GAR, GUZ and SWG were deployed by National Space Science Centre of Chinese Academy of Sciences with instrument EMF-100 (Li, 2022), this type of instrument is independently developed by the Chinese Academy of Sciences, with a range of $\pm50$ kV m$^{-1}$, relative accuracy of $\pm1$ % and resolution of 10 V m$^{-1}$. The LES was deployed by China University of Geosciences (Wuhan) with instrument CS110 (Chen et al.,2021), which has a range of $\pm21.2$ kV m$^{-1}$, a relative accuracy of $\pm1$ %, and a resolution of 3 V m$^{-1}$.

---

## Author Comment (AC2)

Thank you for your valuable comments. We fully acknowledge your suggestions and have revised the paper accordingly. On behalf of all the authors, I would like to respond to you as follows:

**RC2**

**Question 1:**

Observation points are located in the earthquake preparation zone according to the Dobrovolsky formula R=exp(M) (Dobrovolsky et al., 1979). The distance between the observation point and the time between the anomaly and the earthquake is very close to Sidorin's formula log(ΔT×R)=0.72M−0.72, where ΔT is the time in days, and R is the distance in kilometers (Sidorin, 1979).

**Reply:**

Thank you for pointing out the two formulas that significantly enhance the robustness of our article. In reference to Dobrovolsky's formula, we determined the study area's extent by considering the radius derived from this formula. While not explicitly mentioned in the article, we will include this formula in the 'Study Area' section of the manuscript. As for Sidorin's formula, we employed it to conduct certification calculations for the relevant results, leading to improved outcomes. However, we analyzed the pre-seismic anomalies of the Luding EQ in this study, and its results had limited generalizability, hindering accurate validation of the formula. Consequently, we have refrained from citing this reference in this context.

Revised version (Line 91):

The study area was selected as [99°~106° E, 28°~32° N] in consideration of the Dobrovolsky formula (Dobrovolsky et al., 1979) and the geographical locations of the AEF observatories.

Dobrovolsky, I.R., Zubkov, S.I., and Myachkin, V.I.: Estimation of the size of earthquake preparation zones, Pure. Appl. Geophys., 117, 1025-1044, https://doi.org/10.1007/BF00876083, 1979.

**Question 2:**

line 34, to clarify that the upper atmosphere is positively charged.

**Reply:**

The presence of a consistent atmospheric electrostatic field oriented vertically downward during fair weather conditions, along with the existence of opposite charges carried by the atmosphere and Earth, constitutes the cause of the FW-AEF. In accordance with well-established principles of electrical physics, it is acknowledged that electric field lines originate from positive charges and terminate at negative charges.

Revised version (Line 34):

This electric field, also known as the fair-weather atmospheric electric field (FW-AEF), is oriented vertically downwards, which means that the atmosphere is positively charged relative to the Earth, while the Earth carries a negative charge (Li et al., 2022).

Li, L., Chen, T., Ti, S., Wang, S. H., Song, J. J., Cai, C. L., Liu, Y. H., Li, W., and Luo, J.: Fairweather near-surface atmospheric electric field measurements at the Zhongshan Chinese Station in Antarctica. Appl Sci, 12, 9248, https://doi.org/10.3390/app12189248, 2022.

**Question 3:**

line 155, to clarify approximately what time the sun rises and sets in this area according to UTC for the specified period of research.

**Reply:**

Thank you for your valuable suggestions. Upon reviewing the timetable, we have identified that the sunrise time (UTC) within the designated study area during the specified period ranges from 22:41 to 22:50, while the sunset time spans from 11:24 to 11:30. It is crucial to note that all temporal data presented in this paper has been adjusted to local time (LT). Consequently, our manuscript employs local time (LT), signifying that the sunrise occurs between 06:41 and 06:50, with the sunset taking place between 19:24 and 19:30.

---

## Author Comment (AC3)

Thank you for your valuable comments. We fully acknowledge your suggestions and have revised the paper accordingly. On behalf of all the authors, I would like to respond to you as follows:

**RC3:**

**1. Modification of minor points of the manuscript.**
**Point 1:**
Line 9. I think that "four discretely distributed sites" can be simplified in "four sites" as intrinsically "discrete".

**Reply:**
Original text:'recorded at four discretely distributed sites 15d before the Luding EQ'
Revised version: 'recorded at four sites 15d before the Luding EQ'

**Point 2:**
Line 21. The expression "were believed to be" is not so scientific, so I suggest to delete and introduce the sentence in this way: "A possible mechanism of negative AEF anomalies before Luding EQ is supposed that positive charges carriers [...]"

**Reply:**
Original text:'The mechanism of negative AEF anomalies before the Luding EQ were believed to be: positive charge carriers were generated from the underground high stress concentration areas'
Revised version: 'A possible mechanism of negative AEF anomalies before Luding EQ is supposed that positive charge carriers were generated from the underground high stress concentration areas'

**Point 3:**
Line 23 and several occurrences in the manuscript. I think the term "aground" is not proper and I would suggest to change in "above the ground".

**Reply:**
Original text:'thus disturbing the aground AEF'
Revised version: 'thus disturbing the AEF above the ground'
Likewise, occurrences of the word in lines 136, 154, 197, and 340 are replaced with 'above the ground'.

**Point 4:**
Line 30. Instead of "In the state of nature the operation of Global Electric Circuit" you can simply write: "In nature, the Global Electric Circuit"

**Reply:**
Original text: 'In the state of nature, the operation of Global Electric Circuit (GEC) is driven by global thunderstorm activity and large-scale ion separation in charged cloud (Rycroft et al., 2000)'
Revised version: 'In nature, the Global Electric Circuit (GEC) is driven by global thunderstorm activity and large-scale ion separation in charged cloud (Rycroft et al., 2000)'

**Point 5:**

Line 36. Instead of "detected" I suggest to use "claimed" to provide some caution.

Line 37. Please replace "the field mill electric field instrument" by "the field mill instrument" otherwise there are too much repetition of electric field in the same sentence.

**Reply:**

Original text: 'In 1966, Kondon (1966) detected pre-earthquake (EQ) abnormal electric field signal by using the field mill electric field instrument for the first time at the Matsushiro Observatory in Japan'

Revised version: 'In 1966, Kondon (1966) claimed pre-earthquake (EQ) abnormal electric field signal by using the field mill instrument for the first time at the Matsushiro Observatory in Japan'

**Point 6:**

Lines 38-44. Please insert some cautions when report the previous studies, for example at line 43 instead of a typical AEF I suggest to write a possible AEF.

**Reply:**

Original text: 'while the bay-type persistent electric field anomaly monitored at Yongqing station was considered as a typical AEF precursor of the EQ'

Revised version: 'while the bay-type persistent electric field anomaly monitored at Yongqing station was considered as a possible AEF precursor of the EQ'

**Point 7:**

Line 94. I suppose that "observations" stays for "observatories".

**Reply:**

Original text: 'The GAR, GUZ and SWG for AEF observations locate nearby the XSHF while LES locates east to the southwest section of the LQSF'

Revised version: 'The GAR, GUZ and SWG for AEF observatories locate nearby the XSHF while LES locates east to the southwest section of the LQSF'

**Point 8:**

Line 93. Could be worth to mention that Longmen Shan fault was the source of two major earthquakes in last decades, Lushan 2013 and Wenchuan 2008, the last one with catastrophic impact of lives.

**Reply:**

Sentences added to the original text: 'Among these faults, the LMSF was the source of two significant earthquakes in the last few decades: the Lushan EQ in 2013 and the Wenchuan EQ in 2008, the latter having a catastrophic impact on lives.'

**Point 9:**

Line 94. I suppose that "observations" stays for "observatories".

Line 94. You need to introduce the stations/observatories. You can introduce a sentence like: "The atmospheric electric field has been analyzed using four stations called GAR, GUZ, SWG and LES.

**Reply:**

Original text: 'The GAR, GUZ and SWG for AEF observations locate nearby the XSHF while LES locates east to the southwest section of the LQSF'

Revised version: 'The AEF have been analyzed using four observatories called GAR, GUZ, SWG and LES. The first three locate nearby the XSHF while LES locates east to the southwest section of the LQSF.'

**Point 10:**

Table 1. I would suggest to add a column with the altitude of the station after the latitude, even though you reported this information in the text.

**Reply:**

**Table 1. Key information about the AEF observation sites**

| Site name | Longitude | Latitude | Altitude | Distance from the epicenter | Sampling frequency | Unit |
|---|---|---|---|---|---|---|
| GAR | 100.02° E | 31.61° N | 3356 masl | 298.97 km | | |
| GUZ | 102.17° E | 30.12° N | 1421 masl | 59.29 km | 1 s | kV/m |
| SWG | 102.07° E | 29.69° N | 2125 masl | 11.20 km | | |
| LES | 103.91° E | 29.60° N | 401 masl | 175.67 km | | |

**Point 11:**

Line 128. "including but not only", or "among them".

**Reply:**

Original text: 'The AEF is to be influenced by a variety of factors, including not only meteorological factors such as clouds, rain, snow and lightning'

Revised version: 'The AEF is to be influenced by a variety of factors, including but not only meteorological factors such as clouds, rain, snow and lightning'

**Point 12:**

Line 174. The signs of Dst are inverted and it is better insert unit of measurement (nT). A simpler notation would be: "-50 nT < Dst < -30 nT".

**Reply:**

Original text: 'Dst < -50 and > -30 represents weak magnetic storm activity, while SSN > 40 and < 80 represents moderate solar activity'

Revised version: '-50 nT < Dst < -30 nT represents weak magnetic storm activity, while 40 < SSN < 80 represents moderate solar activity'

Similarly, we also added the unit of DST in Figure 3(a).

**Point 13:**

Line 179 (and 180). I think it's more proper to call these data "climatological data" instead of "remote sensing data" even though they are retrieved especially from remote sensing satellite.

**Reply:**

Original text: 'this research conducted a time-series analysis of multiple remote sensing data (CBH, LCC and TP) on an hourly or daily mean basis. For the time period of negative AEF anomalies, remote sensing data was selected to judge whether the non-seismic factors (meteorological parameters) existed synchronously'

Revised version: 'this research conducted a time-series analysis of multiple climatological data (CBH, LCC and TP) on an hourly or daily mean basis. For the time period of negative AEF anomalies, climatological data was selected to judge whether the non-seismic factors (meteorological parameters) existed synchronously'

**Point 14:**

Line 236. I suggest to substitute "geological preference of" with "geological influence on".

**Reply:**

Original text: 'and explained the geological preference of the positive MBT anomaly based on P-hole theory'

Revised version: 'and explained the geological influence on the positive MBT anomaly based on P-hole theory'

**Point 15:**

Line 292 (and 297). Peroxy-defects (not "deficient").

**Reply:**

Original text: 'the production and convergence of P-holes occurs in rocks with peroxy-deficient (peroxy-bonded) structures'

Revised version: 'the production and convergence of P-holes occurs in rocks with peroxy-defects (peroxy-bonded) structures'

Likewise, occurrences of the word in lines 297 is replaced with 'Peroxy-defects'.

**Point 16:**

Line 299. I think "accumulation" would be more proper term than "aggregation".

**Reply:**

Original text: 'it was considered that zones I, II, III and VI are more prone to positive MBT anomalies following P-hole aggregation'

Revised version: 'it was considered that zones I, II, III and VI are more prone to positive MBT anomalies following P-hole accumulation'

**Point 17:**

Line 365. As suggested before please use some caution: "likely had been influenced by seismic activity".

**Reply:**

Original text: 'Therefore, it can be assumed that the negative AEF anomaly at GAR on 24 August had been influenced by seismic activity'

Revised version: 'Therefore, it can be assumed that the negative AEF anomaly at GAR on 24 August likely had been influenced by seismic activity'

**Point 18:**

Line 377. "considered to be possibly associated".

**Reply:**

Original text: 'In conclusion, the negative AEF anomalies at GAR from 13:00 to 15:00 on 24 August and at LES from 18:00 to 19:00 on 22 August were considered to be associated with the surface P-hole accumulation resulted from seismic activity'

Revised version: 'n conclusion, the negative AEF anomalies at GAR from 13:00 to 15:00 on 24 August and at LES from 18:00 to 19:00 on 22 August were considered to be possibly associated with the surface P-hole accumulation resulted from seismic activity'

**2. Specific questions**

**Question 1:**

Line 78. Please, revise the sentence, while you say humidity speaking of atmospheric particles? Maybe some concepts are missing.

**Reply:**

Humidity does not fall within the classification of atmospheric particles. We have replaced the term "atmospheric particles" with "atmospheric parameters" here to avoid conceptual confusion.

**Question 2:**

Line 131. Please, improve the introduction of ERA-5, saying for example that is reanalysis of atmospheric observations from multiple sources (ground, satellite, etc...) produced for climatological investigations.

**Reply:**

Thank you very much for presenting this suggestion, it will certainly help readers gain a deeper understanding of the data product. We once again checked the official introduction to the ERA5 data set and added a paragraph in Line 133: 'ERA5 reanalysis data is a globally complete and consistent data set formed by combining model data with global observation data using the laws of physics, which is widely used for climatological observations.'

**Question 3:**

Lines 148-149. I think that these two sentences can be rephase better even though mostly right for scientific concept. If I understand the meaning I would suggest: "Characterizing the FW-AEF

background is an important, in order to determine the periodic variation of natural AEF in the region under investigation. Consequently, it is necessary [or crucial] to study the characteristics of the FW-AEF to better identify seismic AEF anomalies."

**Reply:**

As mentioned in the previous article, AEF will be affected by many factors. In order to effectively identify earthquake AEF anomalies, it is necessary to study the background characteristic changes of the site's AEF periodic changes (i.e. FW-AEF). Your understanding of this part is quite correct. The version here is modified to: 'In order to ascertain the periodic variations of AEF in the observatory, characterizing the background of FW-AEF is of great importance. Consequently, obtaining a typical AEF curve as the FW-AEF background of GEC is crucial for the identification and extraction of AEF anomalies.'

**Question 4:**

Line 187. I warmly suggest some caution to the authors. Please, change "it had been influenced by seismic activity" in "it had been influenced by seismic activity".

**Reply:**

The modified version you suggested in your question is the same as the original version, but combined with the other modifications you proposed, it should be similar to the problems that appear on lines 365 and 377. The modified version is 'it likely had been influenced by seismic activity'

**Question 5:**

Lines 311-315. I think that also other hypotheses based on difference of electrical propertied of electron with higher mobility than P-holes (separated at source as stated by authors) are possible according to Freund, et al. 2021). Can the authors comments?

**Reply:**

Thank you for your valuable suggestion. In the article (Freund et al., 2021), Freund indeed discussed the activation of charges (electrons and P-holes) in rocks with peroxy-defects when subjected to tectonic stress. According to the P-hole theory, the Earth's crust is considered as a semiconductor. Under the influence of the semiconductor P-N junction, the propagating directions of P-holes and electrons activated by the stress in rocks are quite different. P-holes tend to move along the direction of maximum stress gradient to ground surface, while electrons are to be trapped by P-N junction and do not propagate. Here we also revised this section by incorporating several additional relevant references.

Revised version (Line 311):

The activated P-holes are to flow from the seismic source area to upper crust in response to the direction of maximum stress gradient (Freund et al., 2006, 2021; St-Laurent et al., 2006)

Freund, F., Takeuchi, A., Lau, B. W. S.: Electric currents streaming out of stressed igneous rocks—
        A step towards understanding pre-earthquake low frequency EM emission. Physics and

Chemistry of the Earth Parts A/B/C, 2006, 31, 389-396, https://doi.org/10.1016/j.pce.2006.02.027, 2006.

Freund, F., Ouillon, G., Scoville, J., and Sornette, D.: Earthquake precursors in the light of peroxy defects theory: Critical review of systematic observations. Eur. Phys. J. Spec. Top., 230, 7–46, https://doi.org/10.1140/epjst/e2020-000243-x, 2021.

**Question 6:**

Figure 7. Please insert the vertical scale with unit of measurements and preferable also horizontal one.

**Reply:**

The vertical distance scale consistently remains at 80 kilometers. In order to better illustrate the gradient of ground stress, we have selectively trimmed the original profile, resulting in some variations in the horizontal distance scale. The specific numerical values cannot be ascertained, but given the research area and profile orientation, we hereby present an approximate figure.

[Figure]

**Figure 7.** The vertical sections of crustal stress from the hypocenter or nearby high stress area to the four seismic MBT zones. Subplot (b) and (d) are vertical profiles through the hypocenter, while subplot (a), (c) and (e) are the vertical profile started from the nearby high stress concentration areas.

**Question 7:**

Figure 8 / lines 348-349. Why did you plot the wind field from 3:00 UT on 22 August 2022 and not from 1:00 UT, i.e., before the MBT map at 2:00 UT? Can you add this information, please?

**Reply:**

The sub-figure in Figure 8(a) illustrates an example where fluctuations in the wind field do not lead to abnormal alterations in AEF at the site location. The AEF anomaly at the GAR site occurs from 04:30 to 09:00 on August 22, necessitating the analysis of wind field data at 03:00, 05:00, 07:00, and 09:00. We appreciate your suggestion and have revised the manuscript accordingly, replacing the wind data at 03:00, 05:00, 07:00, and 09:00 with that at 01:00, 04:00, 07:00, and 09:00. In fact, during this time period, the wind field exhibits minimal changes, which better demonstrates the lack of influence from wind field changes on the abnormal AEF at observatories.

[Figure]

**Figure 8(a).** Wind field, MBT residuals and SSM residuals in the study area on 22 August for GAR.

We have thoroughly reviewed the literature you recommended, and we plan to incorporate pertinent citations from these sources in relevant sections of the article:

Line 53:

Freund (2000, 2007, 2010) found that stress-activated carriers, named as P-holes, activated in the igneous and metamorphic rocks, are able to transfer along stress gradient and accumulate on the rock surface in unstressed areas or even on the ground surface covered by sands.

Line 235:

The pre-seismic MBT anomalies in the Qinghai-Tibet Plateau region have been extensively discussed in previous studies. Liu et al. (2023) explored the relationship between MBT anomalies and extensional faults.

Freund, F., Takeuchi, A., Lau, B. W. S., Al-Manaseer, A., Fu, C. C., Bryant, N. A., and Ouzounovet, D.: Stimulated infrared emission from rocks: assessing a stress indicator. eEarth, 2, 7-16, https://hal.science/hal-00298232, 2007.

Liu, S. J., Cui, Y., Wei, L. H., Liu, W. F., and Ji, M. Y.: Pre-earthquake MBT anomalies in the Central and Eastern Qinghai-Tibet Plateau and their association to earthquakes. Remote. Sens. Environ., 298, 113815, https://doi.org/10.1016/j.rse.2023.113815, 2023.